# Andreev reflection of fractional quantum Hall quasiparticles

M. Hashisaka [1,2✉], T. Jonckheere[3], T. Akiho[1], S. Sasaki[1], J. Rech [3], T. Martin[3] & K. Muraki [1]

Electron correlation in a quantum many-body state appears as peculiar scattering behaviour at its boundary, symbolic of which is Andreev reflection at a metal-superconductor interface. Despite being fundamental in nature, dictated by the charge conservation law, however, the process has had no analogues outside the realm of superconductivity so far. Here, we report the observation of an Andreev-like process originating from a topological quantum many-body effect instead of superconductivity. A narrow junction between fractional and integer quantum Hall states shows a two-terminal conductance exceeding that of the constituent fractional state. This remarkable behaviour, while theoretically predicted more than two decades ago but not detected to date, can be interpreted as Andreev reflection of fractionally charged quasiparticles. The observed fractional quantum Hall Andreev reflection provides a fundamental picture that captures microscopic charge dynamics at the boundaries of topological quantum many-body states.

[1] NTT Basic Research Laboratories, NTT Corporation, Atsugi, Kanagawa, Japan. [2] JST, PRESTO, 4-1-8 Honcho, Kawaguchi, Saitama, Japan. [3] Aix Marseille Univ, Université de Toulon, CNRS, CPT, Marseille, France. ✉email: masayuki.hashisaka.wf@hco.ntt.co.jp

When a two-dimensional electron system (2DES) is subjected to a perpendicular magnetic field at low temperatures, electrons condense into the strongly correlated phase of the fractional quantum Hall (FQH) state[1]. Quasiparticles in FQH systems have fascinating properties, such as fractional charge[2] and anyonic statistics[3]. Furthermore, for particular states such as that at Landau-level filling factor $\nu = 5/2$, the theory predicts that quasiparticles obey non-Abelian braiding statistics that provide the basis of fault-tolerant quantum computation[4,5]. The fractional charge[6–9] and anyonic nature[10–12] of the quasiparticles have been revealed experimentally by shot-noise measurements and Fabry–Pérot interferometry. These studies have elucidated the behaviour of quasiparticles within the FQH state—either bulk or edges—that gives their defining properties. On the other hand, one may expect the quasiparticles to exhibit unique behaviour at an interface between the FQH state and another topologically distinct system, in a similar way as the Cooper-pair correlation in a superconductor manifests itself as Andreev reflection, where an electron incident from a normal metal to a superconductor is reflected as a hole[13,14]. This, in turn, poses a fundamental question as to whether electron correlation in a topological quantum many-body state shows up as a unique interface phenomenon. FQH Andreev reflection, which we demonstrate in this paper, is an elementary process that answers this question.

The FQH Andreev process has been predicted by theories examining charge transport across a narrow junction between quantum Hall (QH) states with different filling factors. The most intensively studied system is one comprised of the $\nu = 1/3$ Laughlin state and the $\nu = 1$ integer QH (IQH) state[15,16]. The charge transport can be modelled as the tunnelling between the $\nu = 1/3$ and 1 edge channels, which can be treated as a chiral Luttinger liquid and a Fermi liquid, respectively[17]. When channels are coupled through a single scatterer, the problem can be solved analytically by transforming it into that of tunnelling between edge states with Luttinger parameter $g = 1/2$[18,19]. The exact solution predicts that in the strong-coupling regime the two-terminal conductance $G$ exceeds the conductance $e^2/3h$ ($e$: electron charge, $h$: Planck's constant) of the $\nu = 1/3$ state, reaching $e^2/2h$ in the strong-coupling limit[15,16,18–21]. The enhancement of $G$ can be interpreted as the result of the Andreev process, where two incoming charge-$e/3$ quasiparticles are scattered into a transmitted electron with charge $e$ and a reflected quasihole with charge $-e/3$[15]. This theoretical prediction, however, has not yet been confirmed experimentally, despite recent progress in experiments on related systems[22–26].

In this paper, we present evidence of the FQH Andreev process, namely $G$ exceeding $e^2/3h$ in a narrow junction between $\nu = 1/3$ and 1 states. As the junction width is varied using the split-gate voltage applied to form the junction, $G$ oscillates around $e^2/3h$, exhibiting several peaks where $G$ overshoots the bulk conductance $e^2/3h$, reaching $G \cong 1.2 \times e^2/3h$. The conductance oscillations indicate several Andreev processes at multiple scatterers present between the $\nu = 1/3$ and 1 edges. The evidence is also reinforced by demonstrating that the junction operates as a dc-voltage transformer generating a negative voltage output for positive input.

## Results

**FQH-IQH junction.** Our QH device, formed in a Hall bar containing a 2DES in a GaAs quantum well, has several top gates and pairs of split gates in between (Fig. 1a). A perpendicular magnetic field of $B = 9$ T sets the bulk of the 2DES at $\nu = 1$. We then use the leftmost top gate ($V_L = -0.42$ V) to form a $\nu = 1/3$ region underneath (see the inset in Fig. 2a). A narrow 1/3-1 junction is

formed by applying a negative gate bias $V_S$ to both electrodes of the split gate located immediately to the right of $\nu = 1/3$ region and depleting the 2DES underneath (Fig. 1b). In this situation, the setup for transport measurements can be expressed schematically as in Fig. 1c [see Supplementary Note 1]. We measured the two-terminal differential conductance $dI/dV_{in}$ by applying a source-drain voltage $V_{in} = V_{in}^{dc} + V_{in}^{ac}$ on the $\nu = 1/3$ side of the junction and measuring the transmitted current $I$ on the $\nu = 1$ side using a standard lock-in technique.

**Enhanced two-terminal conductance.** Figure 2a presents the central result of this paper, where we plot the zero-bias conductance $G$, i.e., $dI/dV_{in}$ at $V_{in}^{dc} = 0$ V, as a function of $V_S$. A narrow junction forms at $V_S < -0.55$ V. As $V_S$ is decreased below $-0.55$ V, the junction width decreases and $G$ starts to oscillate around $e^2/3h$ with the amplitude growing with decreasing $V_S$. The most striking observation is that $G$ overshoots $e^2/3h$ at several oscillation peaks before the junction is pinched off at $V_S \cong -1.4$ V. The maximum $G$ reaches $1.2 \times e^2/3h$ at $V_S \cong -1.1$ V. Such a two-terminal conductance, enhanced by narrowing the junction and exceeding the conductance of the constituent element, is nontrivial and counter-intuitive. We note that these features appear only in 1/3-1 junctions and not in 1/3-1/3 or 1-3 junctions (see Supplementary Note 7).

The peculiarity of the charge-transfer process is revealed alternatively by probing the potentials, or voltages $V_i$ ($i = 1$–4) of the incoming and outgoing edge channels. Figure 2b, c displays $V_i$ measured at $V_{in}^{dc} = 0$ V normalised by $V_{in}$, plotted as a function of $V_S$. The voltages $V_1$ and $V_3$ of the incoming channels are, respectively, equal to potentials $V_{in}$ and 0 V of the electrodes on their upstream, independent of $V_S$. In contrast, the voltages $V_2$ and $V_4$ of the outgoing channels vary with $V_S$. The most remarkable feature is the negative voltage that appears in $V_2$. Phenomenologically, this demonstrates that the junction operates as a dc-voltage transformer generating negative voltage output ($V_2 < 0$) for a positive input ($V_{in} > 0$).

From the Landauer–Büttiker formalism, $V_2$ and $V_4$ are related to $G$ as

$$V_2 = [1 - G(e^2/3h)^{-1}]V_{in}, \tag{1}$$

$$V_4 = G(e^2/3h)^{-1}V_{in}. \tag{2}$$

These formulas show that both $V_2 < 0$ and $V_4 > V_{in}/3$ correspond to $G > e^2/3h$. Within the picture of Andreev reflection, the negative voltage ($V_2 < 0$) of the back-reflected channel is a direct manifestation of the quasihole reflection.

**Bias and temperature dependence.** While it is evident that the Andreev reflection is responsible for the observed $G > e^2/3h$, to understand microscopic processes therein, we need to explain the origin of the conductance oscillations, which is not predicted from the original models based on the tunnelling through a single scatterer[18,19]. Resonant tunnelling through unintentional discrete levels in the junction, which are responsible for the oscillations in the low-conductance regime near $V_S = -1.3$ V (see Supplementary Note 4), cannot account for the conductance oscillations with $G > e^2/3h$. In the following, we present the dependence of the conductance on $V_{in}$ and temperature $T$ and discuss the oscillation mechanism.

Figure 3a displays a colour plot of differential conductance $dI/dV_{in}$ as a function of $V_S$ and $V_{in}^{dc}$. The oscillations with $dI/dV_{in} > e^2/3h$ are seen only at $|V_{in}^{dc}| < 40$ μV. For illustration, we plot in Fig. 3b the pinch-off trace at $V_{in}^{dc} = 100$ μV, where $dI/dV_{in} < e^2/3h$ over the entire range of $V_S$. Figure 3c shows the $V_{in}^{dc}$ dependence of $dI/dV_{in}$ at $V_S = -1.113$ and $-0.985$ V. At $V_S = -1.113$ ($-0.985$) V,

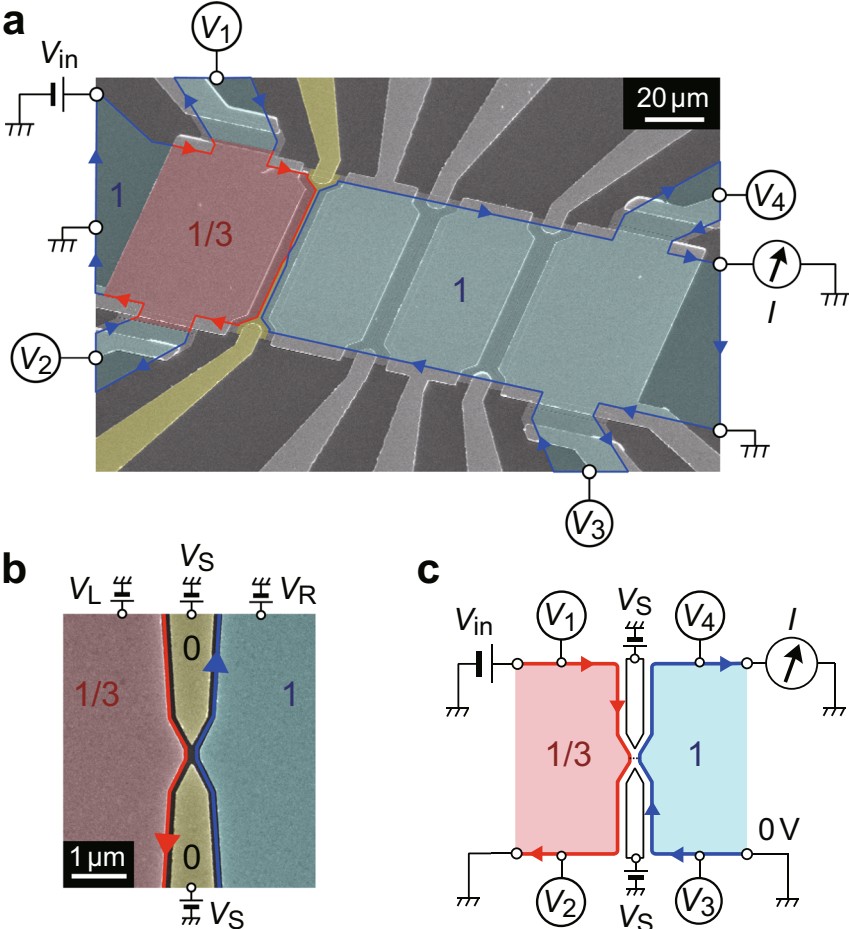

**Fig. 1 Fractional-integer quantum Hall junction. a, b** False-colour scanning electron micrograph of the Hall-bar sample with measurement configurations (**a**) and magnified view near the narrow junction (**b**). A perpendicular magnetic field $B = 9$ T is applied from the back to the front of the sample. The $\nu = 1$ states develop over the wide blue regions in the 2DES with the front-gate voltages, including $V_R$, set at 0 V. Meanwhile, electron density below one of the front gates (red region) is reduced to form the $\nu = 1/3$ state by applying $V_L = -0.42$ V. A narrow 1/3-1 junction is formed by depleting the 2DES under the split gate electrodes (yellow) with negative $V_S$. Chiral edge states are displayed by arrows (blue, between $\nu = 1$ and $\nu = 0$; red, between $\nu = 1/3$ and $\nu = 0$). $V_{in}$ is the applied source-drain voltage, $I$ is the measured current, and $V_i$ ($i = 1$–4) are the measured voltages of the incoming and outgoing channels of the 1/3-1 junction. **c** Schematic of the experimental setup. A narrow junction is formed between $\nu = 1/3$ and $\nu = 1$ states.

which corresponds to the peak (valley) of the oscillations in Fig. 3b, we observe a pronounced zero-bias enhancement (suppression) of the conductance. In contrast, at $V_S = 0$ V, where the $\nu = 1/3$ and 1 regions form a long junction spanning across the 80-μm-wide Hall bar, $dI/dV_{in}$ remains constant at $e^2/3h$. These results clearly show that the Andreev process is observed only in narrow junctions at low bias. The data also reveal that not only the conductance enhancement but also its suppression are low-bias anomalies.

Figure 3d shows the $T$ dependence of the conductance oscillations. The oscillation amplitude decreases with increasing $T$, and the signature of the Andreev process, $G > e^2/3h$, disappears above 200 mK. We focus on two single periods of the oscillations near $V_S = -1.113$ and $-1.095$ V and extract the amplitude $A$ as the peak-to-valley value of $G$ in each period. The two sets of $A$ vs. $T$ data are well fitted by an exponential function $A_0 \exp(-T/T_0)$, as shown in Fig. 3e, where $A_0$ is the amplitude at $T = 0$ and $T_0$ is the characteristic temperature. The exponential temperature dependence bears analogy with that seen in various electronic interferometers[27]. The $T_0$ values (170 and 190 mK) are close to each other, indicating that these oscillations share the same origin in nature. The data in Fig. 3 also demonstrate that the conductance oscillations with $G > e^2/3h$ are highly reproducible (similar conductance oscillations were reproduced for different

cool-downs and in different samples, see Supplementary Notes 3, 5, and 6).

**Multiple-scatterer model**. We argue that several Andreev processes in the junction are responsible for the conductance oscillations, as predicted in theories involving multiple scatterers or a line junction of finite width[18,19,28–32]. We consider $N$ scatterers along the counter-propagating $\nu = 1/3$ and 1 channels and incoherent transport between them. $N$ is proportional to the junction width (i.e., length of the counter-propagating channels) and hence varies with $V_S$. Here, "incoherent transport" means that the $N$ scatterers give independent scattering events, where the outgoing channels are characterised by a chemical potential that defines the input for the next scatterer. With this assumption, the voltage of the $\nu = 1/3$ (1) channel incoming to the $n$th scatterer is given by $V_{n-1} = i_{n-1} \times 3h/e^2$ ($W_{N-n} = j_{N-n} \times h/e^2$), where $i_{n-1}$ ($j_{N-n}$) is the current in the incoming channel (see Supplementary Figure 7). With this setup, one can use the exact solution for the single-scatterer model[19] to define the conductance $g_n$ for each scatterer as a function of the applied bias $V_{n-1} - W_{N-n}$ and evaluate the current $I_n$ flowing through it. Notably, as shown for the single-scatterer case, the charge conservation law and the requirement

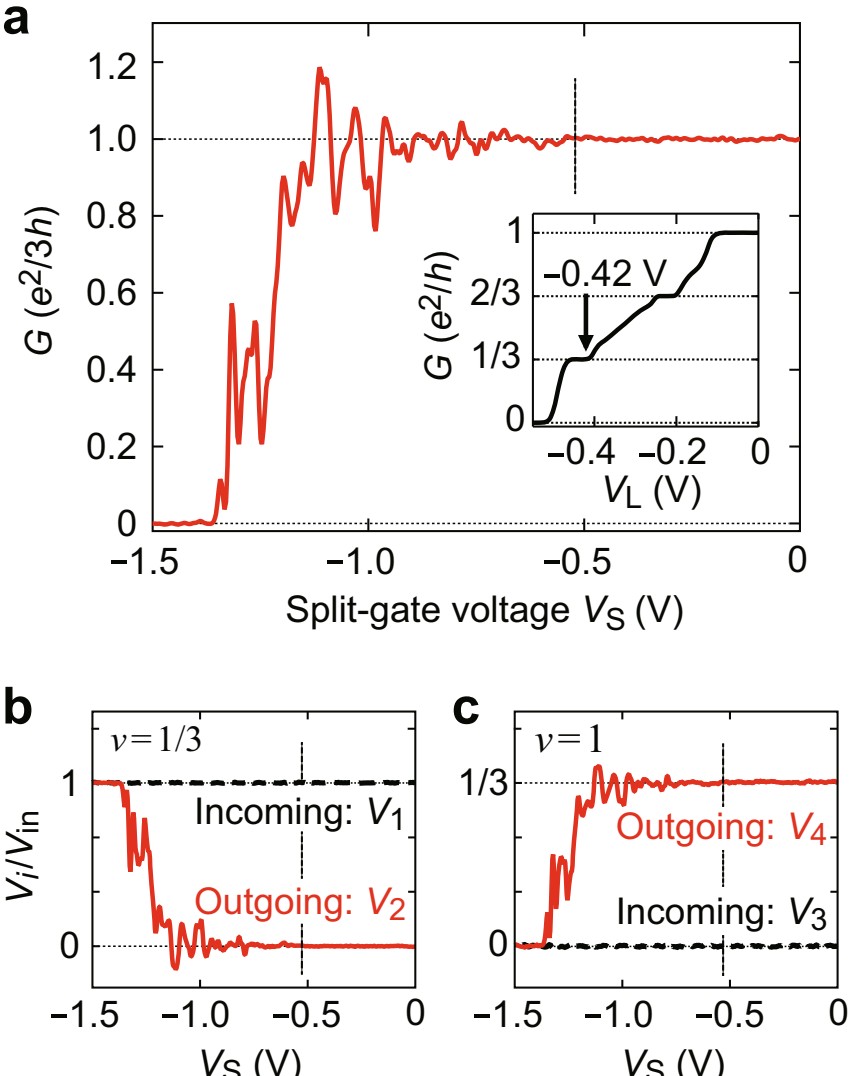

**Fig. 2 Signatures of Andreev reflection. a** Two-terminal conductance $G$ as a function of split-gate voltage $V_S$, taken with $V_L = -0.42$ V. Below $V_S \cong -0.55$ V (indicated by a dashed line), the 2DESs underneath the split gate electrodes are depleted to form a narrow 1/3-1 junction. Conductance oscillations with $G > e^2/3h$, the evidence of the Andreev reflection, are observed. (Inset) $G$ as a function of leftmost top gate voltage $V_L$ measured with $V_S = 0$, indicating that a $v = 1/3$ state forms at $V_L = -0.42$ V in the region immediately to the left of the junction (red region in Fig. 1a, b). **b**, **c** $V_S$ dependence of the voltages on the incoming ($V_1$ and $V_3$) and outgoing ($V_2$ and $V_4$) channels, measured in the same setup as in (**a**) on the $v = 1/3$ (**b**) and $v = 1$ (**c**) sides of the junction. The vertical axes are normalised by the source-drain voltage $V_{in}$. Both negative output ($V_2 < 0$) in the reflected channel and overshoot ($V_4 > V_{in}/3$) in the transmitted channel are the signatures of the Andreev reflection.

that the outgoing power be equal to or less than the incoming one lead to $0 \le g_n \le 1/2$[21]. Here, the charge transport through the $n$th scatterer becomes dissipationless only when $g_n = 0$ or 1/2. The latter (former) corresponds to the strong-coupling limit (complete decoupling). Namely, tunnelling for any intermediate $g_n$ values is accompanied by energy dissipation.

The conductance $G$ of the whole junction is obtained by solving a non-linear system of equations numerically. The results are shown in Fig. 4a for three representative cases: strong ($T_k = 0$, black open circles), intermediate ($T_k = 1.5$ mK, red filled circles), and weak couplings ($T_k = 36$ mK, blue diamonds) under the experimental condition with an applied voltage of 20 μV and a temperature of 9 mK. Here, $T_k$ is the crossover energy scale between strong- and weak-coupling regimes[19] (for details, see Supplementary Note 8). In the strong-coupling limit, where we have $g_n = 1/2$ for all $n$, each scatterer only switches the sign of the

voltage between the channels without causing energy dissipation. Consequently, $G$ oscillates as a function of $N$ between 0 ($N$ even) and $e^2/2h$ ($N$ odd) (black circles). This oscillation can be regarded as the result of successive dissipationless Andreev processes, where the tunnel current switches direction at each scatterer without changing magnitude. When the coupling weakens to give $g_n < 1/2$, each scatterer equilibrates the channels, which results in the reduced output voltage at each scatterer and hence damping of the conductance oscillations (red circles). The damping is significant particularly for large $N$, where the channels experience equilibration many times. When the coupling weakens further to give $g_n < 1/3$ for all $n$, the tunnel current flows only in one direction. In this case, $G$ monotonically increases with $N$, asymptotically approaching $G = e^2/3h$[30] (blue diamonds). The simulation for the intermediate coupling (red circles) captures essential features of the experimental data in Fig. 2a. If we take

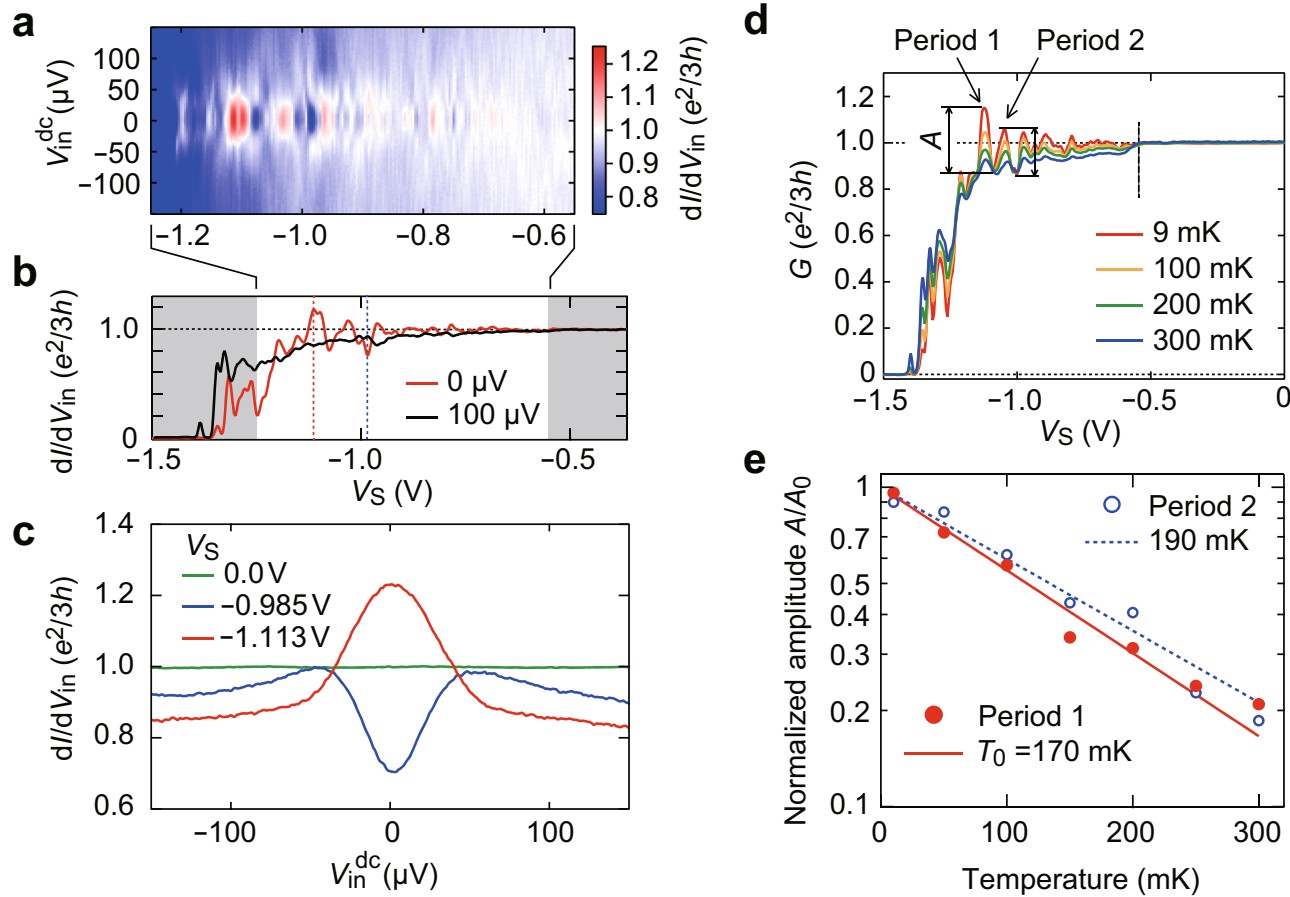

**Fig. 3 Source–drain-voltage and temperature dependence of conductance oscillations. a** Colour plot of $dI/dV_{in}$ as a function of $V_S$ and $V_{in}^{dc}$, indicating oscillations with $dI/dV_{in} > e^2/3h$ at $|V_{in}| < 40$ μV. **b** Comparison of $dI/dV_{in}$ vs. $V_S$ traces at zero and finite bias, indicating that the feature $dI/dV_{in} > e^2/3h$ is absent at $V_{in}^{dc} = 100$ μV. **c** $V_{in}^{dc}$ dependence of $dI/dV_{in}$ at several $V_S$. Zero-bias conductance enhancement (suppression) is observed at $V_S = -1.113$ V ($-0.985$ V), whereas no feature is seen without forming a narrow junction ($V_S = 0$ V). **d** $G$ vs. $V_S$ traces measured at several temperatures. The amplitudes $A$ of the oscillations in periods 1 and 2 are estimated as the peak-to-valley values. **e** Temperature dependence of $A/A_0$ for periods 1 and 2. The curves of the form $\exp(-T/T_0)$ well fit the data using similar characteristic temperatures ($T_0 = 170$ and $190$ mK).

into account more realistic experimental situations, including the confining potential of the split gate and randomness in the positions of the scatterers, the simulation can even better reproduce the experimental features (Fig. 4b). In the simulation, the confining potential controls the effective width of the junction by multiplying a position-dependent window function to the coupling strength of the scatterers, resulting in weaker coupling near the junction ends and hence reduced oscillation amplitude (for details, see Supplementary Note 8).

While the above multiple-scatterer model well explains the $V_S$ dependence of $G$, it still fails to account for the observed $V_{in}$ and $T$ dependence. Since the model inherits the $V_{in}$ and $T$ dependence of the conductance from the single-scatterer model[18,19], which gives $dI/dV_{in}$ as an increasing function of $V_{in}$ and $T$, it remains incapable of reproducing oscillations decaying with $V_{in}$ or $T$. This, in turn, suggests that coherent processes neglected in the above model, such as interference between successive scattering events[33], play an important role. We speculate that constructive (destructive) interference of tunnelling amplitudes can enhance (suppress) the coupling strengths of several scatterers in some range of $V_S$. Indeed, a theory considering coherent interference predicts that a 1/3-1 junction with Coulomb interaction shows conductance oscillations up to $e^2/2h$ as a function of the junction width[28]. In this view, conductance enhancement or suppression at the extrema of the oscillations is partly due to the interference

enhancement of the coupling. This picture explains why the oscillations appear only at low $V_{in}$ and $T$. Furthermore, it helps to understand why the simulation for the weak-coupling regime of the incoherent model can mimic the $V_S$ dependence of $G$ at high $V_{in}$ (Fig. 3b) or high $T$ (Fig. 3d).

## Discussion

Finally, we discuss a related interesting issue, namely the mixing of the $\nu = 1/3$ and 1 edge modes expected for a wide junction. Counter-propagating $\nu = 1/3$ and 1 channels studied here is a basic setup in the model for the edge modes of the hole-conjugate $\nu = 2/3$ FQH state[34]. There, inter-channel Coulomb interaction and disorder-assisted tunnelling govern the mixing of the channels and thus determine their fate in the low-temperature and long-channel limit. The conductance oscillations with $G$ exceeding $e^2/3h$ observed in our experiment can be interpreted as a precursor phenomenon of the mixing process, namely the "mesoscopic fluctuation" predicted in ref. [32], suggesting the presence of the neutral-mode physics of counter-propagating channels at the 1/3-1 junction[31,32]. Our findings indicate that the Andreev process is a vital ingredient therein.

We have demonstrated FQH Andreev reflection, which is one of the essential concepts for understanding edge transport at the boundaries of topological quantum many-body systems. We

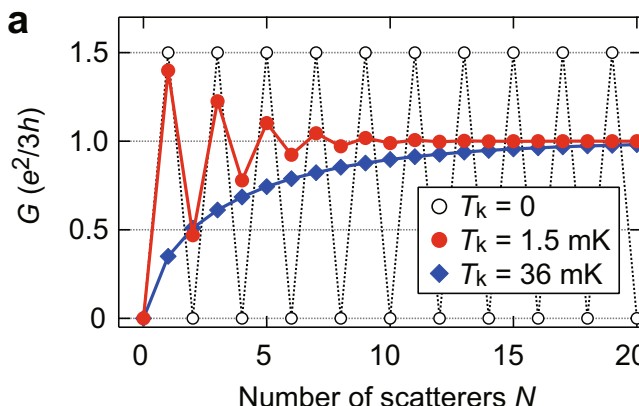

**a**

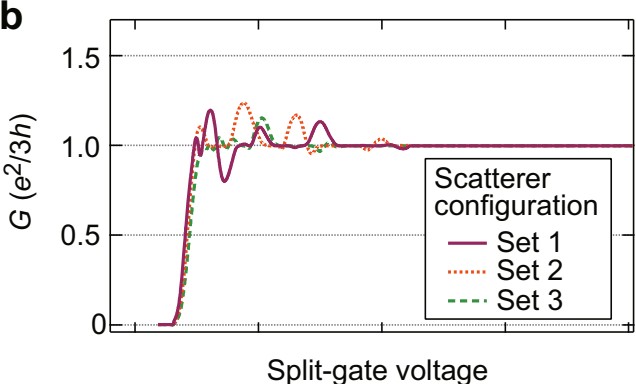

**b**

**Fig. 4 Simulation of conductance oscillations using incoherent *N*-scatterer model. a** $N$ dependence of $G$ for several coupling strengths: strong-coupling limit ($T_k = 0$, black open circles), intermediate ($T_k = 1.5$ mK, red filled circles), and weak couplings ($T_k = 36$ mK, blue diamonds). **b** Simulations taking the effects of the confining potential and the randomness in the positions of the scatterers into account. A window function was used to relate the split-gate voltage with the number and strength of scatterers (for details, see Supplementary Note 8). The three traces (1–3) are obtained for different configurations of the scatterers' positions.

expect to observe similar Andreev processes in various FQH junctions with different electronic systems, including non-QH systems such as normal metals and superconductors[35,36].

## Methods

**Sample fabrication.** We fabricated the sample in a 2DES in a GaAs quantum well of 30 nm width. The centre of the well is located 190 nm below the surface. The sample was patterned using e-beam lithography for fine gate structures and photolithography for chemical etching, coarse metalized structures, and ohmic contacts formed by alloying Au–Ge–Ni on the surface.

**Measurement setup.** We set electron density in the 2DES at $2.2 \times 10^{11}$ cm$^{-2}$ by applying a back-gate voltage of 1.29 V at a refrigerator temperature of 9 mK, except for the data in Fig. 3d, e. A perpendicular magnetic field $B = 9$ T was applied from back to the front of the sample. The lock-in measurements were performed with the ac modulation of $V_\text{in}^\text{ac} = 20$ µV RMS at 31 Hz. The experimental results demonstrated in the main text were obtained for the split-gate device with an opening of 300 nm. The data from the devices with wider apertures are available in Supplementary Note 6.

## Data availability

All data that support the plots within this paper and other findings of this study are available from the corresponding author upon reasonable request.

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

## Acknowledgements

The authors thank T. Ito, N. Shibata, and T. Fujisawa for fruitful discussions and H. Murofushi for technical support. This work was supported by Grants-in-Aid for Scientific Research (Grant nos. JP16H06009, JP15H05854, JP26247051, and JP19H05603) and JST PRESTO Grant no. JP17940407.

## Author contributions

M.H. conceived the experiment. M.H., T.A., and S.S. fabricated the sample. M.H. performed the measurement and analysed the data. T.J. and K.M. performed the simulations. M.H., T.J., and K.M. interpreted the results with help from J.R. and T.M. M.H. wrote the paper with help from T.J. and K.M. All authors discussed the results and commented on the paper.

## Competing interests

The authors declare no competing interests.
