## [Peer Review File · Nature Communications]

REVIEWER COMMENTS

Reviewer #1 (Remarks to the Author):

The manuscript "Andreev reflection of fractional quantum Hall quasiparticles" by Hashisaka et al. observes that in a narrow junction between fractional and integer quantum Hall edge states, the two-terminal conductance when biasing the fractional state exceeds that of the constituent fractional state. This observation can be understood in terms of Andreev reflection of fractionally charged particles.

The subject of the paper is of interest and the experimental data is convincing. I recommend publication in nature communications.

As pointed out in the end of the manuscript, there was a concrete theoretical study for disordered edges with counter-propagating $\nu = 1$ and $\nu = 1/3$ (Ref. 28). When the disorder-induced tunneling is relevant in the renormalization group sense, the system may be driven to a disorder-dominated phase, characterized by a "mesoscopic fluctuation" of the conductance. The conductance oscillates depending on the specific disorder realization. Is this analysis compatible with the present experimental data? The experimental observation that the oscillation is suppressed in T and V_{in} seems in a good agreement with the analysis; the renormalization stops at T or V_{in} before reaching the disorder-dominated phase.

Minor comments:

It would be instructive to mention a few recent experimental progresses on making the $\nu = 1$ and $\nu = 1/3$ junction:

Phys. Rev. Lett. 113, 266803 (2014); Nature Communications 10, 1920 (2019); Phys. Rev. Lett. 125, 076802 (2020)

In Fig. 4, it may be better to put a unit of $k_B T_k$ in Kelvin instead of Joule.

Reviewer #2 (Remarks to the Author):

I have no problem with the manuscript, as the data is fine
the effect is clear, but quite small

Yet, to be definite about the claims, one has to measure the charge and find out if indeed it is that of an electron on the $\nu=1$ and $e/3$ on the $\nu=1/3$ side – in the regime that AR is claimed to take place

the paper though is hard to follow – e.g., the captions, that are expected to teach the reader when he/she looks at the data, say nearly nothing

This is just one example, that suggests that better editing is needed

Reviewer #3 (Remarks to the Author):

The manuscript from Hashisaka et al. presents the Andreev-like process by coupling $\nu=1/3$ and $\nu=1$ quantum Hall edges through a quantum point contact structure. By tuning the split-gate voltage, a conductance oscillation is observed around the constituent value of $e^2/3h$. This behavior is attributed by authors the quasiparticle tunneling resembling an Andreev process which is expected from a

normal-superconductor interface. By using fractional and integer quantum Hall edge states, the authors can reproduce the Andreev-like process without a superconductor. The bias and temperature results were further analyzed. Lastly, this unpredicted oscillating behavior in conductance is addressed by the multiple scattering process which depends on the coupling strength, detail of the QPC potential, and the number of scatters.

The manuscript provides a clear signature of coupling between two quantum ground states through the gate control. From bias and temperature dependence, the authors demonstrated the Andreev-like process. This work indeed provides the first observation of the Andreev-like process based on quasiparticle reflection between fraction-integer quantum Hall states. Previous work from I. P. Radu et al. *Science* 320, 899 (2008) has discussed the quasiparticle tunneling from $\nu=5/2$ state. A recent result on the bilayer graphene system also explored the interesting physics on coupling fractional quantum Hall states (*Nat. Phys.* 15, pages 893–897 (2019)). By hybridizing these exotic quantum states, emerging new physics can lead to an unconventional transport behavior. Although the paper seems to fulfill the requirements to be published in *Nature Communications*, I would like to clarify a few points from the manuscript.

1. The bias was applied from the $\nu=1/3$ side. Have authors check the symmetry-bias or bias from the $\nu=1$ side. By reversing the voltage biasing direction, will it provide similar results to the main observation? Yet another possibility is to change the gate setting, therefore two states can be swapped. This may provide more evidence for this quasiparticle transport.

2. One signature of the Andreev process is the particle-hole symmetry due to superconductivity. Although the authors provide a bias study in the manuscript, it may be important to discuss the symmetry from this quasiparticle charge transfer procedure.

3. Continue with the bias spectrum, a characteristic energy scale of 40 μeV has been measured. Often, this energy in the Andreev process can be correlated to the superconducting gap or energy of the Andreev bound states. Could authors comment on this energy scale of 40 μeV ? How could readers understand it? Also, the activation energies from two oscillations were measured to be 170 mK and 190 mK which corresponding to 15 to 16 μeV . How do we understand these values with respect to the result from the bias spectrum?

4. In figure 2a, a pronounced oscillation is observed. However, this oscillation is continued to -1.3 V before the channel is completely pinched off. Similar temperature dependence can be also found for this low conductance oscillation. Based on the numerical simulation, the lower conductance ($G < e^2/3h$) oscillation is not predicted. Could authors comment on this point? Or discuss possible mechanisms?

5. Following the previous point, in the lower conductance region based on figure 3 b ($V_s = -1.2$ to -1.4V), it seems to suggest that there is only dip behavior for the bias dependence. The 100 μV curve is always higher than 0 μV . This provides a different behavior compared to $V_s = -1.2$ to -1.0V . Is there any intuition about this difference?

6. The simulation provides some insight into the quasiparticle transport process. However, it doesn't provide a good agreement with experimental results. For example, the authors claim that intermedia coupling, the red curve in fig 4b, captures the experimental results the best. However, the red curve provides mostly enhancement in conductance with very little suppression. Instead, the black dot curve seems to resemble the result from the experiment, isn't it?

7. Compared to the numerical result, the oscillation amplitude decays relatively fast when the QPC channel becomes wider. On the other hand, the magnitude of oscillation doesn't reach $1.5 e^2/3h$

even for the dissipationless scattering case, e.g. the strong coupling scenario. It was not discussed in the main text the cause of this discrepancy. A small discussion could help readers to understand it better.

8. It was not stated clearly, but the split gate V_s is applied simultaneously. In SI, the V_{s1} and V_{s2} are applied separately as shown in fig. S3. In this case, a few oscillating peaks or dips can be found discretely distributed. These oscillations do not show a clear parallel behavior as mentioned in the SI. In fact, there seem to have more oscillations by cutting along V_{s1} than V_{s2} . In the same SI section, it is not clear to me that the claim of a discrete level cannot induce $G > e^2/3h$. Could the authors elaborate a bit more?

9. From the field dependence, fig. S4, indeed it seems to rule out the interference effect such as AB interference. One expects the same period in the field due to AB interference. However, it was not completely clear to me why these peaks only exist in a small range of magnetic fields?

10. After reading the main text, I found the link between SI and the main text is not great. It may help readers a lot to point out which section of SI that they should look into. A few questions after reading the main text can be resolved by SI. However, it is not easy without going through SI. For example, other scenarios with both $\nu=1/3$ or $\nu=1$ in two regions were presented in SI. However, it was not mentioned clearly in the main text. Additional samples were shown in SI without a word in the main text. By making a proper link from the main text to SI, I believe that readers can understand this work better.

Reviewer #1

The manuscript “Andreev reflection of fractional quantum Hall quasiparticles” by Hashisaka et al. observes that in a narrow junction between fractional and integer quantum Hall edge states, the two-terminal conductance when biasing the fractional state exceeds that of the constituent fractional state. This observation can be understood in terms of Andreev reflection of fractionally charged particles.

The subject of the paper is of interest and the experimental data is convincing. I recommend publication in nature communications.

[Reply]

We thank reviewer 1 for his/her high evaluation of our work and recommendation for publication in Nature Communications. Below we would like to address the issues raised by reviewer 1.

[Comment 1]

As pointed out in the end of the manuscript, there was a concrete theoretical study for disordered edges with counter-propagating $\nu = 1$ and $\nu = 1/3$ (Ref. 28). When the disorder induced tunneling is relevant in the renormalization group sense, the system may be driven to a disorder-dominated phase, characterized by a “mesoscopic fluctuation” of the conductance. The conductance oscillates depending on the specific disorder realization. Is this analysis compatible with the present experimental data? The experimental observation that the oscillation is suppressed in T and V_{in} seems in a good agreement with the analysis; the renormalization stops at T or V_{in} before reaching the disorder-dominated phase.

[Reply 1]

Yes, we consider that the conductance oscillations around $G = e^2/3h$ observed in our experiment correspond to the “mesoscopic fluctuation” in the renormalization group (RG) theory. The theory predicts that at low temperatures two-terminal conductance $G_{2/3}$ of a small $\nu = 2/3$ FQH system takes any value between $G_{2/3} = e^2/3h$ and $4e^2/3h$ depending on the realization of disorder (Ref. 28, renumbered Ref. 32 in the revised manuscript). The values $G_{2/3} = e^2/3h$ and $4e^2/3h$ correspond, respectively, to $G = e^2/2h$ and 0 of the conductance between counter-propagating $\nu = 1/3$ and 1 channels at the $\nu = 2/3$ edge, namely to the enhanced conductance in the strong-coupling limit in the present Andreev-reflection picture and completely decoupled $\nu = 1/3$ and 1 channels. Our experimental results show conductance oscillations around $G = e^2/3h$ as a function of split-gate voltage V_S , where

varying V_S corresponds to changing the system length and disorder realization. The observed suppression of the conductance oscillations at high temperatures can be interpreted as a result of the decrease in the equilibration length between the counter-propagating channels, which is consistent with the RG theory picture. While these issues are interesting, in-depth discussion requires more experiments. We, therefore, leave it as future work. We expect to observe charge-neutral-mode physics, which is predicted by the theory for counter-propagating $\nu = 1$ and $\nu = 1/3$ channels, if we measure heat transport along with the 1/3-1 interface, as pointed out in Ref. 28 (Ref. 32 in the revised manuscript). We modified the relevant sentence to make clear the link between our experiment and the theory.

[Comment 2]

It would be instructive to mention a few recent experimental progresses on making the $\nu = 1$ and $\nu = 1/3$ junction: Phys. Rev. Lett. 113, 266803 (2014); Nature Communications 10, 1920 (2019); Phys. Rev. Lett. 125, 076802 (2020).

[Reply 2]

We thank reviewer 1 for this comment. We described the recent experiments on $\nu = 1$ and $\nu = 1/3$ junctions in the main text and added these papers in the references.

[Comment 3]

In Fig. 4, it may be better to put a unit of $k_B T_k$ in Kelvin instead of Joule.

[Reply 3]

We agree with reviewer 1 in this point. We revised the manuscript following this comment.

We thank the reviewer again for the valuable inputs. It is of great pleasure if the reviewer finds the above responses and revisions appropriate.

Reviewer #2

I have no problem with the manuscript, as the data is fine the effect is clear, but quite small.

[Reply]

We thank reviewer 2 for his/her high evaluation of our work. Below we would like to address the issues raised by reviewer 2.

[Comment 1]

Yet, to be definite about the claims, one has to measure the charge and find out if indeed it is that of an electron on the $\nu=1$ and $e/3$ on the $\nu=1/3$ side; in the regime that AR is claimed to take place.

[Reply 1]

The enhancement of the dc conductance demonstrated in our manuscript is evidence of the Andreev-like reflection at the $1/3-1$ junction, in the same way as that enhanced conductance through a superconductor-normal metal junction is regarded as evidence of the standard Andreev reflection. On the other hand, as pointed out by reviewer 2, it is important to measure the charge of the excitations on both sides of the junction for gaining deeper insight into the microscopic dynamics of charge carriers in the Andreev process. We would like to consider this as a future issue.

[Comment 2]

the paper though is hard to follow – e.g., the captions, that are expected to teach the reader when he/she looks at the data, say nearly nothing

This is just one example, that suggests that better editing is needed

[Reply 2]

Following this comment, we revised the manuscript, particularly the figure captions. We believe that readability is much improved.

We thank the reviewer again for the valuable inputs. It is of great pleasure if the reviewer finds the above responses and revisions appropriate.

Reviewer #3

The manuscript from Hashisaka et al. presents the Andreev-like process by coupling $\nu=1/3$ and $\nu=1$ quantum Hall edges through a quantum point contact structure. By tuning the split-gate voltage, a conductance oscillation is observed around the constituent value of $e^2/3h$. This behavior is attributed by authors the quasiparticle tunneling resembling an Andreev process which is expected from a normal-superconductor interface. By using fractional and integer quantum Hall edge states, the authors can reproduce the Andreev-like process without a superconductor. The bias and temperature results were further analyzed. Lastly, this unpredicted oscillating behavior in conductance is addressed by the multiple scattering process which depends on the coupling strength, detail of the QPC potential, and the number of scatters.

The manuscript provides a clear signature of coupling between two quantum ground states through the gate control. From bias and temperature dependence, the authors demonstrated the Andreev like process. This work indeed provides the first observation of the Andreev-like process based on quasiparticle reflection between fraction-integer quantum Hall states. Previous work from I. P. Radu et al. Science 320, 899 (2008) has discussed the quasiparticle tunneling from $\nu=5/2$ state. A recent result on the bilayer graphene system also explored the interesting physics on coupling fractional quantum Hall states (Nat. Phys. 15, pages 893-897(2019)). By hybridizing these exotic quantum states, emerging new physics can lead to an unconventional transport behavior. Although the paper seems to fulfill the requirements to be published in Nature Communications, I would like to clarify a few points from the manuscript.

[Reply]

We thank reviewer 3 for his/her thorough reading and high evaluation of our work. Below we would like to address the issues raised by reviewer 3.

[Comment 1]

The bias was applied from the $\nu=1/3$ side. Have authors check the symmetry-bias or bias from the $\nu=1$ side. By reversing the voltage biasing direction, will it provide similar results to the main observation? Yet another possibility is to change the gate setting, therefore two states can be swapped. This may provide more evidence for this quasiparticle transport.

[Reply 1]

While we always applied the ac modulation voltage V_{ac}^{in} to the $\nu = 1/3$ side on the left of the

junction, the bias-direction dependence can be known from the colour plot of the differential conductance dI/dV_{in} in Fig. 3a. In two-terminal conductance measurements, reversing the polarity of the dc bias V_{dc}^{in} gives the same results as that for reversing the bias direction. The behaviour of dI/dV_{in} , which is symmetric with respect to the V_{dc}^{in} polarity, clearly shows that the same Andreev-reflection features are observed if we apply V_{ac}^{in} on the $\nu = 1$ side.

[Comment 2]

One signature of the Andreev process is the particle-hole symmetry due to superconductivity. Although the authors provide a bias study in the manuscript, it may be important to discuss the symmetry from this quasiparticle charge transfer procedure.

[Reply 2]

While the process we observe can be referred to as “Andreev reflection” since it causes the reflection of quasiholes, the physics at work is quite different from that of a normal-metal/superconductor (N/S) junction. Therefore, at least now, we cannot provide a rigorous discussion of the particle-hole symmetry in the “Andreev reflection” of the fractional quasiparticles. In the N/S junction, the Andreev reflection is caused by the energy gap and the condensate of Cooper pairs. In our FQH-IQH junction, there is no gap for the edge states, but instead, the different filling factors cause the peculiar charge reflection. The basic process is that two $e/3$ quasiparticles incident on the junction are reflected as a quasihole with the transmission of an electron because only charge e can travel in the $\nu = 1$ side. The constraint imposed by the filling factors can be seen in the maximum value of the conductance, which is $1.5e^2/3h$ for our system, as discussed in N. Sandler *et al.*, PRB **57**, 12324 (1998) (Ref. 15).

[Comment 3]

Continue with the bias spectrum, a characteristic energy scale of 40 μeV has been measured. Often, this energy in the Andreev process can be correlated to the superconducting gap or energy of the Andreev bound states. Could authors comment on this energy scale of 40 μeV ? How could readers understand it? Also, the activation energies from two oscillations were measured to be 170 mK and 190 mK which corresponding to 15 to 16 μeV . How do we understand these values with respect to the result from the bias spectrum?

[Reply 3]

We consider that the observed energy scale of 40 μeV is related to the coupling strength between the $\nu = 1/3$ and 1 edge channels. The three simulated traces in Fig. 4a demonstrate the coupling-strength dependence of the oscillation amplitude at fixed bias and temperature.

This result indicates that the oscillation is suppressed when T_k , the crossover energy scale between strong and weak coupling regimes (Ref. 19), becomes higher (in other words, the coupling becomes weaker) than the applied bias or temperature. We added descriptions in the main text to clarify this point. Although our experiments suggest that the finite bias effectively weakens the coupling (i.e., increases T_k) by suppressing coherent processes (see discussions in the main text), as the precise mechanism of the bias affecting the coherence is not known at present, one cannot quantitatively relate the bias and T_k .

As reviewer 3 pointed out, the threshold voltage of 40 uV at which the conductance oscillations are terminated does not precisely correspond to the characteristic temperature estimated from the temperature dependence. In the present experiment, the conductance oscillations result from the Andreev processes at multiple scatterers in the 1/3-1 junction. In this case, in contrast to the case of electron interferometers having fixed interference paths, the bias and temperature dependence can differ since the bias may vary the microscopic charge dynamics. On the other hand, the fact that the observed energy scales are of the same order suggests that such bias effect is not so significant in the present case. We thank reviewer 3 for bringing this point to our attention.

[Comment 4]

In figure 2a, a pronounced oscillation is observed. However, this oscillation is continued to ≈ 1.3 V before the channel is completely pinched off. Similar temperature dependence can be also found for this low conductance oscillation. Based on the numerical simulation, the lower conductance ($G < e^2/3h$) oscillation is not predicted. Could authors comment on this point? Or discuss possible mechanisms?

[Reply 4]

The oscillations in the low-conductance regime near $V_g = -1.3$ V do not result from Andreev processes but from resonant tunnelling through discrete levels unintentionally formed near the junction. As discussed in section 4 of SI, the nature of the conductance oscillations can be distinguished by looking at their behaviour when asymmetric voltages (V_{S1} and V_{S2}) are applied to the split gate. The peaks in the low-conductance regime shift diagonally (Fig. S3b), in contrast to those in the high-conductance regime moving parallel to either V_{S1} or V_{S2} (Fig. S3a). We added a sentence describing this point in the main text.

[Comment 5]

Following the previous point, in the lower conductance region based on figure 3 b ($V_s = \approx 1.2$ to ≈ 1.4 V), it seems to suggest that there is only dip behavior for the bias dependence. The

100 uV curve is always higher than 0 uV. This provides a different behavior compared to $V_s = 1.2$ to 1.0 V. Is there any intuition about this difference?

[Reply 5]

As explained in **[Reply4]**, the peaks in the low-conductance regime near $V_g = -1.3$ V originate from resonant tunnelling through unintentionally formed discrete levels. The observation that the conductance increases monotonically with bias or temperature in this regime is consistent with this resonant-tunnelling picture.

[Comment 6]

The simulation provides some insight into the quasiparticle transport process. However, it doesn't provide a good agreement with experimental results. For example, the authors claim that intermedia coupling, the red curve in fig 4b, captures the experimental results the best. However, the red curve provides mostly enhancement in conductance with very little suppression. Instead, the black dot curve seems to resemble the result from the experiment, isn't it?

[Reply 6]

In the corresponding text, we refer to the red circles in Fig. 4a, not in Fig. 4b. We have added a note in the main text to clarify this point and changed the colour of the traces in Fig. 4b.

The three curves in Fig. 4b were calculated assuming the same coupling strength but different realisations of scatterers. The purpose of this simulation is not to discuss which trace resembles the experimental data but to examine the influence of the confining potential and the positions of the scatterers. Please see **[Reply 7]** for more details.

[Comment 7]

Compared to the numerical result, the oscillation amplitude decays relatively fast when the QPC channel becomes wider. On the other hand, the magnitude of oscillation doesn't reach $1.5 e^2/3h$ even for the dissipationless scattering case, e.g. the strong coupling scenario. It was not discussed in the main text the cause of this discrepancy. A small discussion could help readers to understand it better.

[Reply 7]

Indeed, we have not observed a conductance reaching $1.5 \times e^2/3h$, the maximum possible value. The reduced conductance can be understood as due to the confining potential from the split gate; that is, negative V_s not only squeezes the constriction but also tends to decrease the

coupling strength of the scatterers, which makes it difficult to reach the few-scatterer regime while retaining the strong coupling. In other words, our FQH-IQH junction always includes some scatterers with reduced coupling. This can also be seen in the theoretical results shown in Fig. 4b: although each scatterer has a quite strong bare coupling (that would individually give a conductance close to $1.5 \times e^2/3h$), because of the confining potential, the conductance of the whole junction shows oscillations with peak values much reduced from $1.5 \times e^2/3h$. We have added a sentence in the manuscript explaining this point.

[Comment 8]

It was not stated clearly, but the split gate V_s is applied simultaneously. In SI, the V_{s1} and V_{s2} are applied separately as shown in fig. S3. In this case, a few oscillating peaks or dips can be found discretely distributed. These oscillations do not show a clear parallel behavior as mentioned in the SI. In fact, there seem to have more oscillations by cutting along V_{s1} than V_{s2} . In the same SI section, it is not clear to me that the claim of a discrete level cannot induce $G > e^2/3h$. Could the authors elaborate a bit more?

[Reply 8]

We added a description to the main text that V_s is applied simultaneously to both electrodes of the split gate.

In Fig. S3a, some peaks and dips around $e^2/3h$ appear in parallel to V_{s1} or V_{s2} , e.g., peaks (red regions) at $V_{s1} \cong -0.8$ V. As pointed out by the reviewer, these features may not have been clearly visible in the previous manuscript. We revised Fig. S3a tweaking the range of the colour map. We believe that it is clearer in the revised figure that some oscillations are arranged in parallel to V_{s2} as well (particularly at $V_{s1} \cong -0.8$ V).

Resonant tunnelling through a discrete level can explain the oscillations of the conductance but cannot its overshoot $G > e^2/3h$. In an FQH-IQH junction, discrete levels can form between the macroscopic $\nu = 1/3$ and $\nu = 1$ regions that serve as the leads to the junction. Because the $\nu = 1/3$ lead has the conductance $e^2/3h$, without Andreev process, the two-terminal conductance of the whole device cannot exceed $e^2/3h$, even in the case of resonance-induced perfect transmission (transmission probability $T = 1$) through the discrete level. To make this point clearer, we added a description in section 4 of SI.

[Comment 9]

From the field dependence, fig. S4, indeed it seems to rule out the interference effect such as AB interference. One expects the same period in the field due to AB interference. However, it

was not completely clear to me why these peaks only exist in a small range of magnetic fields?

[Reply 9]

In the field-dependent data shown in Fig. S4a, AB-interference-like oscillations are seen only in the limited gate-voltage range around $V_S = -1.113$ V. As the reviewer notes, this may suggest that the main features of the conductance oscillations with $G > e^2/3h$ (shown in Fig. 2a) do not require interference effect. We note, however, that varying the magnetic field affects both the coupling strength and the spatial arrangement of the junction. Therefore, we believe Fig. S4a to be useful (though not entirely convincing) supplementary data suggestive of the interference effect involved. The reason why some peaks disappear outside the small range of the fields can be understood as due to the changes in the realisation of multiple scatterers.

[Comment 10]

After reading the main text, I found the link between SI and the main text is not great. It may help readers a lot to point out which section of SI that they should look into. A few questions after reading the main text can be resolved by SI. However, it is not easy without going through SI. For example, other scenarios with both $\nu=1/3$ or $\nu=1$ in two regions were presented in SI. However, it was not mentioned clearly in the main text. Additional samples were shown in SI without a word in the main text. By making a proper link from the main text to SI, I believe that readers can understand this work better.

[Reply 10]

We thank reviewer 3 for this comment. Following the comment, we added section numbers of SI in the revised manuscript. We also added a few sentences to make links between the main text and SI. We believe that these revisions improve the readability of the manuscript.

We thank the reviewer again for the valuable inputs. It is of great pleasure if the reviewer finds the above responses and revisions appropriate.

REVIEWERS' COMMENTS

Reviewer #1 (Remarks to the Author):

In the revised version the authors have adequately answered my questions on the relation between “mesoscopic conductance fluctuation” and Andreev reflection. Even though they did not reach the maximum possible conductance value of $e^2/2h$, the present work is worth reporting. I recommend the publication of the manuscript in Nature Communications.

Reviewer #3 (Remarks to the Author):

After reading the response and revision of the manuscript, I am satisfied with the authors' reply. Last few comments on the manuscript.

1. The V_s has a different style in the caption of figure 2. The subscript S is italic.
2. I will suggest having a legend for figure 4.

Reviewer #1

In the revised version the authors have adequately answered my questions on the relation between “mesoscopic conductance fluctuation” and Andreev reflection. Even though they did not reach the maximum possible conductance value of $e^2/(2h)$, the present work is worth reporting. I recommend the publication of the manuscript in Nature Communications.

[Reply]

We thank reviewer 1 for his/her recommendation for publication in Nature Communications.

Reviewer #3

After reading the response and revision of the manuscript, I am satisfied with the authors' reply. Last few comments on the manuscript.

1. The V_s has a different style in the caption of figure 2. The subscript S is italic.
2. I will suggest having a legend for figure 4.

[Reply]

We thank reviewer 3 again for his/her high evaluation of our manuscript. Following the reviewer's comments, we changed the subscript S of V_s to roman (caption of Fig. 2) and added figure legends for Figures 4(a) and 4(b) in the revised manuscript.